# Retention of volunteers and factors influencing program performance of the Senior Care Volunteers Training Program in Jiangsu, China

**Hong-Li Chen**[1☯], **Pei Chen**[1☯], **Yu Zhang**[1], **Ying Xing**[1], **Yuan-Yuan Guan**[1], **Dao-Xiang Cheng**[2], **Xian-Wen Li**[1]*

1 School of Nursing, Nanjing Medical University, Nanjing, Jiangsu, China, 2 Red Cross Society of Jiangsu Branch, Nanjing, Jiangsu, China

☯ These authors contributed equally to this work.
* xwli0201@njmu.edu.cn

**Data Availability Statement:** All relevant data are within the manuscript and its Supporting Information files.

## Abstract

### Background

As the country with the largest aging population, China faces an enormous challenge with its elderly support and care. One of the proposed solutions is the development of volunteerism for elderly care. The Senior Care Volunteers Training Program (SCVTP) was initiated by the Red Cross Society of China with the purpose of training volunteers to care for community seniors. As one of the four pilot provinces, Jiangsu Province launched the program since 2017.

### Aims

The present study was conducted to investigate the dropout rate of trained volunteer group leaders, the characteristics of the retained trained volunteer group leaders and the activities that their groups conducted. Additionally, the exploration of the factors influencing the SCVTP's performance was listed as another aim.

### Methods

A cross-sectional study was designed. The study used purposive sampling to select participants who meet the criteria from all the trained volunteer group leaders (n = 623). Demographic questionnaire, volunteer role identity (VRI) scale, attitude toward helping others (AHO) scale, team climate and atmosphere (TCA) scale, and volunteer program performance evaluation (VPPE) questionnaire were used to collect the data online. Descriptive statistics were used to determine the dropout rate and general characteristics of the retained volunteers and the activities. A multiple linear regression equation was developed to study the factors that influence program performance.

**Funding:** This work was supported by the Project of Philosophy and Social Science Research in Colleges and Universities in Jiangsu Province (2017SJB0285). The funders had no role in study design, data collection and analysis, decision to publish, or preparation of the manuscript.

**Competing interests:** The authors have declared that no competing interests exist.

## Results

In total, 307 questionnaires were valid in the study. About 67.9%, 53.7%, and 30.0% of the trained volunteer group leaders dropped out of the program in the year of 2017, 2018, and 2019, respectively. The retained trained volunteer group leaders were more likely to be females (84.7%), those in excellent health (75.2%) and with a bachelor's degree or above (87.6%). Less attention has been paid to frailty care (n = 76) than other volunteer caring activities (e.g., safe care: n = 277, diet care: n = 250, drug management care: n = 226). VRI (β = 0.118, p = 0.017), AHO (β = 0.134, p = 0.021), TCA (β = 0.459, p<0.001), and financial sustainability (β = 0.179, p<0.001) affected the SCVTP's performance significantly (adjusted $R^2$ = 0.356).

## Conclusion

High rate of trained volunteer group leaders' dropout should be brought to the policymaker's attention. The characteristics of the retained trained volunteer group leaders provide a useful reference for the recruitment of trainees in the future. Frailty care may need more training by the volunteer service provider. In order to enhance program performance, a better team climate and atmosphere, financial sustainability, and volunteers with appropriate attitude and role identity are also necessary for the volunteer program.

## Introduction

Population aging is a global demographic megatrend [1]. According to World Population Prospects 2019, 16.7% of people will be aged 65 years or over by 2050 worldwide [2]. As the country with the largest population, China has approximately 176 million persons aged 65 years or over, accounting for 12.6% of the country's total population by the end of 2019 [3]. Compared with developed countries, the status quo of "getting old before getting rich" in China makes it likely to face more fiscal pressures in relation to public systems of healthcare, pensions and social protection schemes for older persons [1]. Previous studies on elderly care services indicated that volunteers can supplement family members' care and paid staff care [4–7]. The participation of trained volunteers in old-age care could also be beneficial for the delivery of caring knowledge to relatives and paid housekeepers of community seniors [8]. In addition, the care for seniors provided by volunteers can relieve the physiological and psychological care pressure on caregivers or paid housekeepers [9].

To make use of volunteer resources to cope with the aging society, the Red Cross Society of China initiated the first national volunteer training program for community senior care, named the Senior Care Volunteers Training Program (SCVTP), in 2017 [10]. As one of the four pilot provinces, Red Cross Society of Jiangsu Branch has been developing the program since 2017. For the program, the volunteer group leaders with at least 2 years of experience in senior care were recommended by the local Red Cross Society. A two-stage training were designed in the program. The first stage is the training of the trainer (ToT), which aimed to culture and improve the caring ability of the volunteer group leaders, and the second stage was expected to educate volunteer groups and to supply volunteer care for the local community seniors. For the first stage, a 7-day ToT course was utilized, including lecture and skill practice. The course was developed based on brainstorming between eight nursing professors, two Red Cross administrators, one medicine education professor, and one human medicine doctor. If

the volunteer group leaders could pass the test, he or she will be afforded a certificate. There is no payment for them. However, Red Cross Society of Jiangsu Branch supplied 20 thousand Chinese Yuan to encourage trained volunteer group leaders to launch a new volunteer group or develop an original group taking care for local community senior.

Evaluation is a critical part of volunteer training programs to clarify the advantages and problems of the project and to take more targeted measures for promoting the quality and sustainable development of the project [11]. Retaining volunteers has proved a challenge for volunteer organizations. Previous studies found that high turnover causes the costly training of replacement volunteers and lowers the productivity and morale for the organization [12,13]. Considering these factors, the first aim of the present study is to investigate the dropout of ToT-trained volunteer group leaders and the second aim is to describe the characteristics of retained volunteer group leaders and the activities that their groups have completed. Many factors can affect volunteer program performance directly. Besides, some factors can affect volunteer participation or personal performance and thus affect program performance. The factors mentioned above include team factors such as financial sustainability [14,15], team climate and atmosphere [16–20], and personal factors such as gender [21,22], education background [31,32], retirement status [23], attitude toward helping others [24–27], volunteer role identity [28–31]. Based on the above literature review, monitoring of the factors and intervention are necessary in this dynamic environment to improve program performance [32–34]. Therefore, the third aim of this paper is to explore the factors influencing volunteer program performance. The Project Monitoring and Evaluation Guide, designed by the International Federation of the Red Cross and Red Crescent, was regarded as a useful tool for volunteers and partners involved in development programs [35]. The guide was used as the evaluation framework in the present study.

## Methods

### Ethical considerations

This study was approved by the Ethics Committee of Nanjing Medical University, Red Cross Society of Jiangsu Branch and School of Nursing, Nanjing Medical University. Before the survey, the aims and the main contents of the study were briefly described to the participants. They were told they could refuse the survey and had the right to withdraw from the study at any stage. Electronic informed consent was obtained from each participant. The opportunity to participate in consolidation training in the next step and the financial support from the Red Cross Society of Jiangsu Branch were not influenced even if the volunteers refused to participate in this study.

### Study design

A population-based, cross-sectional design was utilized in this study.

### Study participants and dropout

All the volunteer group leaders who attended SCVTP developed by Red Cross Society of Jiangsu Branch in the year of 2017, 2018 and 2019 were recruited to take part in this study. According to the record, 623 volunteer group leaders attended this program. The sampling frame in the study consisted of a roster of all the trained volunteer group leaders. Previous research has shown that purposive sampling method may prove to be effective when only limited numbers of people can serve as primary data sources due to the nature of research design and aims [36]. The study used purposive sampling to select participants who met the criteria,

which helped in simplifying the selection of the subject of the investigation. The criteria used to measure retainment of participants are as follows: (a) The trained volunteer group leaders who did not respond to this investigate were defined as one kind of dropout. (b) Even though the trained volunteer group leaders responded to this study but did not launch the volunteer group program carrying out activities related to community senior care services (e.g., diet care, respite care, chronic disease care) were defined as one kind of dropout. (c) The respondents who did not meet the criterion of average response time for each question of spending at least 2 seconds on each question [37] or who self-reported that they answered questions carelessly [38] were defined as the dropout.

## Instruments

**The demographic questionnaire.** The demographic questionnaire included gender, age, job condition and other characteristics of the ToT volunteer group leaders. Volunteer team status, such as team size, activities time, average hours of one activity, types of activities (e.g., diet care, safe care, excretion care), and finance sustainability, were also investigated. The options in "types of activities" characteristic are multiple choices.

**Volunteer Role Identity scale (VRI).** The VRI is a five-item scale designed by Grube and Piliavin [39]. The items are as follows: (a) My volunteer work is something I rarely ever think about; (b) I would feel a loss if I were forced to give up volunteering; (c) I really don't have any clear feelings about volunteer work; (d) For me, being a volunteer means more than just doing volunteer work; and (e) Volunteering is an important part of who I am. The answer categories ranged from (1) strongly disagree to (7) strongly agree. The higher scores of the VRI, the more important the volunteer identity to them. The items formed a reliable scale in Grube and Piliavin's study ($\alpha = 0.82$). The Cronbach's $\alpha$ of VRI in the present study was 0.543.

**Attitude Toward Helping Others scale (AHO).** The AHO is a 4-item instrument measuring the degree of helping attitudes [40]. The authors define the AHO as "global and relatively enduring evaluations with regard to helping or assisting other people." The items are as follows: (a) People should be willing to help others who are less fortunate; (b) Helping troubled people with their problems is very important to me; (c) People should be more charitable toward others in society; and (d) People in need should receive support from others. Respondents answered each item using a 5-point Likert scale ranging from 1 (strongly disagree) to 5 (strongly agree). Higher AHO scores indicate a stronger willingness to help others. The four items for the AHO in the two samples had acceptable Cronbach's $\alpha$ of 0.79 and 0.80, respectively. The Cronbach's $\alpha$ of the AHO in this study was 0.834.

**Team Climate and Atmosphere scale (TCA).** The TCA [41] has 10 items, which are mainly about trust, support, feelings, care among team members, recognition of individual performance, efficiency of solving conflicts, freeness of expressing opinion, and individual influence on team decisions. The TCA is part of The Long-Term Care version of Team STEPPS 2.0 that reflect the environment of nursing homes and other long-term care settings, such as assisted living and continuing care retirement communities. The answer categories ranged from (1) strongly disagree to (5) strongly agree. The higher scores of the scale are, the more harmonious the team atmosphere is. The Cronbach's $\alpha$ of the scale in the present study was 0.824.

**Volunteer Program Performance Evaluation (VPPE) questionnaire.** The questionnaire [42] was designed based on the Project Monitoring and Evaluation Guide via a two-round Delphi method to measure the program performance. After the discussion of our research team, efficiency (time and activity efficiency), effectiveness (effectiveness for the elderly and their family), relevance (relevance for the elderly), impact (impact on volunteer teams and the

elderly) and sustainability (program and activity plan sustainability) were kept to the evaluation questionnaire. As the degree of satisfaction is regarded as one of the vital factors of care [43]. The degree of satisfaction (satisfaction of the elderly and their family about the program) was added. The concordance coefficients of the questionnaire ranged from 0.163 to 0.318 and 0.162 to 0.222. and the variation coefficients ranged from 0.081 to 0.245 and 0.059 to 0.204 for the two rounds. The Cronbach's α of the scale in the present study was 0.982, indicating good reliability of the tool. The answer categories ranged from 0 to 10, and the score was weighted according to the tool description. The VPPE score was the dependent variable. Higher weighted score indicates the better program performance. The development process and some details of the two-round Delphi method have been introduced in one Chinese article written by our research group.

## Data collection

A self-rated formative evaluation was conducted in this study. The evaluation questionnaire was delivered to all trained volunteer group leaders in 2017, 2018 and 2019 via twelve *WeChat* groups, an online communication tool widely used in China. To increase the response rates, we employed incentives in our study ranging from 1 to 10 RMB, which was lottery-based where the respondent had a chance to win the incentive [44]. The participants were informed that they could receive the incentive before they answered the questionnaires. Three reminders were also sent to volunteer group leaders about the online questionnaire. Data were collected in late December (between 27 and 31) 2019 only once.

## Statistical analysis

The software IBM® SPSS® Statistics (Armonk, New York, USA) was used to analyze the data. Descriptive statistics were used to determine the general characteristics of the subjects. The comparison of the VPPE scores between variables of trained volunteers and volunteer teams was tested by independent-sample t test and one-way ANOVA, and the correlation between variables was examined by bivariate correlation. Multiple linear regression analysis was performed to study the variables that can significantly influence volunteer program performance.

## Results

### Demographic description

In total, 310 trained volunteers submitted the questionnaires. According to the standard (response time and self-reported diligence), there was one respondent who spent less than 2 seconds on average answering each question and 2 respondents who reported that they did not answer the questions carefully or seriously. In total, 307 questionnaires were valid.

According to the record from the Red Cross Society of Jiangsu Branch, 215 volunteers in 2017, 201 volunteers in 2018, and 207 volunteers in 2019 attended the ToT program. A total of 67.9% of ToT volunteer group leaders trained in 2017 dropped out, and 53.7% and 30.0% dropped out in 2018 and 2019, respectively. The total dropout rate of all the ToT volunteer group leaders is 49.3%. The ratios between the three years were significantly different (P<0.001). (Table 1).

According to Table 2, the mean age of the retained volunteer group leaders was 41.96 years (Standard Deviation: 10.24). Approximately nine-tenths of them (281, 91.5%) were already married. Three-fifths of the respondents (231, 75.2%) were in excellent health. The educational level of the majority of the volunteer group leaders was the equivalent of a bachelor's degree (183, 59.6%).

Table 1. The number of respondents and trained volunteers in 2017, 2018, and 2019.

| Training Year | Retained volunteers, n(%) | Total (N) | $\chi^2$ | p |
|---|---|---|---|---|
| 2017 | 69(32.2) | 215 | 61.859 | <0.001 |
| 2018 | 93(46.3) | 201 | | |
| 2019 | 145(70.0) | 207 | | |

## Team and activity description

Table 3 reveals the characteristics of retained volunteer group leaders' teams and their activities. The size of most teams was less than 29 volunteers. Approximately two-thirds of the groups' activities for the community senior were carried out no more than 60 times since they set up, while a quarter were carried out more than 100 times. The average hours of one activity for most teams varied from 1 to 2 hours. More than half of the volunteer teams (56.4%) reported that they did not have enough money to support the development of the program. As for the details of the activity contents, most volunteer groups (n>200) supplied more safe care, diet care, drug management care, and psychological care for the community senior, while less frailty care was supplied. Compared with the three years of 2017, 2018 and 2019, there was no statistically significant differences in different team sizes, different activities times and some other characteristics in the table.

Table 2. Characteristics of retained volunteer group leaders (N = 307).

| Characteristics | Options | Frequency (n) | Percentage (%) |
|---|---|---|---|
| *Gender* | Female | 260 | 84.7 |
| | Male | 47 | 15.3 |
| *Age* (years, $\bar{x}\pm s$: 41.96±10.24) | <45 | 186 | 60.6 |
| | 45~ | 106 | 34.5 |
| | ≥60 | 15 | 4.9 |
| *Marital status* | Married | 281 | 91.5 |
| | Others | 26 | 8.5 |
| *Health status* | Excellent | 231 | 75.2 |
| | Well | 68 | 22.1 |
| | General | 8 | 2.6 |
| *Care for the elderly at home* | Yes | 178 | 58.0 |
| | No | 129 | 42.0 |
| *Translation into parenthood* | Yes | 276 | 89.9 |
| | No | 31 | 9.1 |
| *Education level* | Below high school | 9 | 2.9 |
| | High school | 29 | 9.4 |
| | Undergraduate | 183 | 59.6 |
| | Above Undergraduate | 86 | 28.0 |
| *Employment* | Employed | 277 | 90.2 |
| | Retired | 30 | 9.8 |
| *Occupation* | Doctors | 26 | 8.5 |
| | Nurses | 161 | 52.4 |
| | Preventive medicine workers | 11 | 3.6 |
| | Social workers | 70 | 22.8 |
| | Others | 39 | 12.7 |

## Descriptive statistics of the main variables

The average scores of the VRI ($\bar{x}$ = 5.89), AHO ($\bar{x}$ = 4.66), TCA ($\bar{x}$ = 4.42) and VPPE ($\bar{x}$ = 8.49) were near the end of the possible maximum. It is clear from the standard deviation (s = 1.01, 0.51, 0.51, 1.33) that there was no considerable variation around the mean. (Table 4).

## Comparison of the VPPE scores between different ToT volunteer leaders and volunteer teams

It can be seen from Table 5 that among different characteristics of the volunteer group leaders (gender, age, marital status, etc), there were no differences in the VPPE scores, while the VPPE scores were significantly different in different team size and finance.

## Correlation between VRI, AHO, TCA and VPPE

VRI, AHO, and TCA were all positively correlated with the VPPE scores, and all the correlation coefficients were statistically significant. (Table 6).

## Multiple linear regression analysis of the VPPE scores

The variables of VRI ($\beta$ = 0.118, p = 0.017) AHO ($\beta$ = 0.134, p = 0.021), TCA ($\beta$ = 0.459, p<0.001), and finance ($\beta$ = 0.179, p<0.001) affected the SCVTP's performance which were

**Table 3. Characteristics of retained volunteer group leaders' teams and activities, N(%).**

| Characteristics | Options(n = 307) | 2017(n = 69) | 2018(n = 93) | 2019(n = 145) | Z/χ² | p |
|---|---|---|---|---|---|---|
| *Team size (persons)* | <15, 116(37.8) | 20(29.0) | 38(40.9) | 58(40.0) | 0.547 | 0.761 |
| | 15~29, 89(29.0) | 26(37.7) | 26(28.0) | 37(25.5) | | |
| | 30~44, 42(13.7) | 16(23.2) | 12(12.9) | 14(9.7) | | |
| | 45~89, 27(8.8) | 4(5.8) | 8(8.6) | 15(10.3) | | |
| | ≥90, 33(10.7) | 3(4.3) | 9(9.7) | 21(14.5) | | |
| *Activities times* | <20, 61(19.9) | 9(13.0) | 22(23.7) | 30(20.7) | 0.013 | 0.994 |
| | 20~39, 81(26.4) | 25(36.2) | 17(18.3) | 39(26.9) | | |
| | 40~59, 58(18.9) | 14(20.3) | 20(21.5) | 24(16.6) | | |
| | 60~79, 22(7.2) | 3(4.3) | 8(8.6) | 11(7.6) | | |
| | 80~99, 11(3.6) | 3(4.3) | 4(4.3) | 4(2.8) | | |
| | ≥100, 74(24.1) | 15(21.7) | 22(23.7) | 37(25.5) | | |
| *Average hours of one activity* | <1, 16(5.2) | 2(2.9) | 5(5.4) | 9(6.2) | 1.459 | 0.482 |
| | 1~, 158(51.5) | 39(56.5) | 42(45.2) | 77(53.1) | | |
| | 2~, 82(26.7) | 21(30.4) | 29(31.2) | 32(22.1) | | |
| | ≥3, 51(16.6) | 7(10.1) | 17(18.3) | 27(18.6) | | |
| *Finance* | Inadequate, 173(56.4) | 41(59.4) | 46(49.5) | 86(59.3) | 2.566 | 0.277 |
| | Adequate, 134(43.6) | 28(40.6) | 47(50.5) | 59(40.7) | | |
| *Types of respite care (multiple choice)* | Safe care, 277(90.2) | 63(91.3) | 84(90.3) | 130(84.4) | 7.056 | 0.996 |
| | Diet care, 250(81.4) | 61(88.4) | 73(78.5) | 116(75.3) | | |
| | Drug manage care, 226(73.6) | 59(85.5) | 63(67.7) | 104(67.5) | | |
| | Excretion care, 163(53.1) | 41(59.4) | 46(49.5) | 76(49.4) | | |
| | Clean care, 198(64.5) | 48(69.6) | 68(73.1) | 82(53.2) | | |
| | Sleep care, 155(50.5) | 39(56.5) | 46(49.5) | 70(45.5) | | |
| | Rehabilitation Care, 146(47.6) | 37(53.6) | 44(47.3) | 65(42.2) | | |
| | Frailty care, 76(24.8) | 21(30.4) | 17(18.3) | 38(24.7) | | |
| | Chronic disease care, 186(60.6) | 43(62.3) | 53(57.0) | 90(58.4) | | |
| | Psychological care, 218(70.0) | 53(76.8) | 61(65.6) | 104(67.5) | | |
| | Others, 59(19.2) | 17(24.6) | 15(16.1) | 27(17.5) | | |

**Table 4. Descriptive statistics of the main variables.**

| Variables | α | $\bar{x}$ | s | Range | Possible range |
|---|---|---|---|---|---|
| *VRI* | 0.543 | 5.89 | 1.01 | 3~7 | 1~7 |
| *AHO* | 0.834 | 4.66 | 0.51 | 1.25~5 | 1~5 |
| *TCA* | 0.824 | 4.42 | 0.51 | 1.4~5 | 1~5 |
| *VPPE* | 0.982 | 8.49 | 1.33 | 0.52~10.00 | 0~10.00 |
| Relevance | 0.941 | 2.81 | 0.43 | 0.17~3.19 | 0~3.19 |
| Efficiency | 0.760 | 0.28 | 0.07 | 0.00~0.37 | 0~0.37 |
| Effectiveness | 0.962 | 0.87 | 0.17 | 0.01~1.08 | 0~1.08 |
| Satisfaction | 0.981 | 1.33 | 0.19 | 0.11~1.48 | 0~1.48 |
| Social impact | 0.929 | 0.58 | 0.11 | 0.03~0.70 | 0~0.70 |
| Sustainability | 0.955 | 2.60 | 0.56 | 0.20~3.19 | 0~3.19 |

VRI: volunteer role identity; AHO: attitude toward helping others; TCA: team climate and atmosphere; VPPE: volunteer project performance evaluation; Range: minimum to maximum score of the scale; Possible range: actual minimum to maximum score that respondents answered.

significant at the 0.05 level. A unit change in VRI, AHO, TCA, finance increased program performance by 0.118, 0.134, 0.459, 0.179 units, respectively. The adjusted $R^2$ of the model (0.356) revealed that 35.6 percent of variance in participation could be explained by the four aforementioned variables. (Table 7).

## Discussion

The present study found that close to half (49.3%) of trained volunteer leaders dropped out of the program in total (67.9%, 53.7%, and 30.0% for 2017, 2018, and 2019, respectively). For the training of first year, that is, 2017, nearly seventy percent of trainees dropped out according to the operative definition of this study, which is beyond the expectation of the policymaker. Although dropout trends, including multi-measurements, for 2017 and 2018 were not acquired as no panel data were available in this study, the dropout ratio comparison of the three years could also imply that the dropout rate increased over time, which is the first main finding of the present study. Osamu Watanabe and Jiraporn Chompikul conducted a study in terms of turnover intention of village health volunteers (VHVs) who engaged in elderly home care in a Thailand volunteer program and found that approximately one-third of the VHVs intended to leave volunteering in elderly care [45]. To a certain extent, the actual dropout rate or the turnover intention of such a training program is normal. The high dropout rate of the present program, however, should be brought to the forefront of the program design and to the policymaker's attention. Further study to explore the reasons that result in a high turnover rate is needed, as large amounts of time and money have been invested to train the volunteers.

The present study investigated the characteristics of retained trained volunteer leaders. They were more likely to be female and married, in excellent health and with a bachelor's degree. A previous study demonstrated that the majority of the volunteer work force is comprised of women and highly educated individuals [21]. Some findings stated that retirement also plays an important role in volunteers. The retirement effect is accounted for by investing retirees' extra time in the volunteer role, signaling a compensation strategy [23]. The characteristics of the retained trained volunteer leaders provide a useful reference for the recruitment of trainees in the future. For the retained volunteer leaders' groups and the activities, most of the teams in this study were small- to medium-sized. One to two hours was the normal time for general activities. Volunteer groups supplied more care concerning safe care, diet care and

**Table 5. Comparison of the VPPE scores between different ToT volunteer leaders and volunteer teams.**

| Characteristics | Options | $\bar{x}$ | s | t/F | p |
|---|---|---|---|---|---|
| *Gender* | Female | 8.53 | 1.28 | 1.405 | 0.161 |
| | Male | 8.23 | 1.56 | | |
| *Age* | <45 | 8.56 | 1.21 | 0.722 | 0.487 |
| | 45~ | 8.37 | 1.49 | | |
| | ≥60 | 8.39 | 1.53 | | |
| *Marital status* | Married | 8.47 | 1.35 | -0.607 | 0.544 |
| | Others | 8.64 | 0.99 | | |
| *Health* | Excellent | 8.54 | 1.25 | 1.788 | 0.169 |
| | Well | 8.41 | 1.27 | | |
| | General | 7.67 | 3.08 | | |
| *Care for the elderly at home* | Yes | 8.59 | 1.23 | 1.662 | 0.098 |
| | No | 8.34 | 1.34 | | |
| *Translation into parenthood* | Yes | 8.47 | 1.34 | -0.701 | 0.484 |
| | No | 8.65 | 1.22 | | |
| *Education* | Below high school | 8.16 | 3.01 | 1.426 | 0.235 |
| | High school | 8.91 | 0.82 | | |
| | Bachelor | 8.49 | 1.32 | | |
| | Above bachelor | 8.37 | 1.20 | | |
| *Employment* | Employed | 8.48 | 1.34 | -0.176 | 0.860 |
| | Retired | 8.53 | 1.23 | | |
| *Position* | Doctors | 8.75 | 0.90 | 1.385 | 0.239 |
| | Nurses | 8.48 | 1.27 | | |
| | Preventive medicine workers | 8.96 | 0.97 | | |
| | Social workers | 8.53 | 1.33 | | |
| | Others | 8.12 | 1.77 | | |
| *Team size* | <15 | 8.14 | 1.41 | 4.191 | 0.003 |
| | 15~29 | 8.65 | 1.17 | | |
| | 30~44 | 8.72 | 1.07 | | |
| | 45~89 | 8.45 | 1.82 | | |
| | ≥90 | 9.03 | 0.96 | | |
| *Activities times* | <20 | 8.25 | 1.46 | 1.956 | 0.085 |
| | 20~39 | 8.42 | 1.33 | | |
| | 40~59 | 8.37 | 1.54 | | |
| | 60~79 | 8.51 | 1.19 | | |
| | 80~99 | 8.22 | 1.74 | | |
| | ≥100 | 8.88 | 0.89 | | |
| *Average hours of one activity* | <1 | 8.49 | 1.30 | 1.285 | 0.280 |
| | 1~ | 8.36 | 1.30 | | |
| | 2~ | 8.57 | 1.15 | | |
| | ≥3 | 8.75 | 1.63 | | |
| *Finance* | Inadequate | 8.33 | 1.33 | 2.303 | 0.022 |
| | Adequate | 8.68 | 1.30 | | |

psychological care for community seniors, which may have something to do with the curricula related to these fields delivered to the volunteers. Frailty care has received little attention in the implementation of activities, though it is also vital to community seniors. One study provided evidence that frailty is associated with increased healthcare costs, requires more attention in

**Table 6. Correlation coefficient between VRI, AHO, TCA and VPPE (r/p).**

|  | VPPE | VRI | AHO | TCA |
|---|---|---|---|---|
| VPPE | 1.000 | \ | \ | \ |
| VRI | 0.289** | 1.000 | \ | \ |
| AHO | 0.426** | 0.317** | 1.000 | \ |
| TCA | 0.544** | 0.247** | 0.575** | 1.000 |

** Correlation is significant at the 0.01 level; VRI: volunteer role identity; AHO: attitude toward helping others; TCA: team climate and atmosphere; VPPE: volunteer project performance evaluation.

aging societies and is a major task for healthcare systems [46]. Given this, the volunteer service provider may need more training in frailty care.

The third main finding of the study is that team climate and atmosphere were the principal factors contributing to volunteer program performance. Previous research has shown that team climate positively affects relational coordination and communication between team members [47], and there is a positive relationship between team climate and members' satisfaction [48–50]. It was also examined in an earlier study that inter-team coordination and personal satisfaction with the job or organization are conducive to team effectiveness [51,52]. First, considering the relationships among the TCA, teamwork, satisfaction and program performance, making more efforts to support a positive team climate by the leader of the volunteer team, such as creating harmony team culture and delivering it to the new members when they participate, is the basis for improving the performance of the organization and program. Second, the coordination and communication between members are influenced by team climate, so it is believed to promote program development by enhancing the quality of underlying relationships. To attain these aims, all the committed efforts directed toward a highly interdependent common goal can make team members closely interact and coordinate their efforts, which is important to program performance, as collective efficacy can be solidified [53]. Last but not least, regarding the level of personal satisfaction, in addition to the efforts made to support the team climate, attention should be paid to other strategies related to incentives, support and commitment.

The study also revealed that volunteer role identity, attitude toward helping others, and finance were factors contributing to volunteer program performance. Volunteer role identification could make volunteers feel proud to be members and devote themselves to common goals. Teams are cohesive when volunteer participants identify with their membership, and the result of joint efforts is further conducive to program performance [54]. A team member with a positive attitude can improve individual performance, leading to holistic development in the performance of the organization [24,55]. It is the same with attitude toward helping

**Table 7. Coefficients of the variables for the VPPE score regression equation.**

|  | B | S.E. | β | p | 95% CI |
|---|---|---|---|---|---|
| (Constant) | -0.015 | 0.671 | / | 0.982 | [-1.335,1.305] |
| VRI | 0.155 | 0.065 | 0.118 | 0.017 | [0.028,0.282] |
| AHO | 0.346 | 0.149 | 0.134 | 0.021 | [0.053,0.638] |
| TCA | 1.197 | 0.148 | 0.459 | <0.001 | [0.906,1.487] |
| Finance | 0.478 | 0.124 | 0.179 | <0.001 | [0.233,0.723] |

Dependent Variable: VPPE; VRI: volunteer role identity; AHO: attitude toward helping others; TCA: team climate and atmosphere; B: unstandardized regression coefficient; S.E.: Std.Error; β: standardized regression coefficient; p: p-value; CI: confidence interval.

others in the area of volunteerism. To meet these expectations, organizations must establish systematic human resource management and select appropriate volunteers to meet organizational objectives, which can lead organizations to develop better. In regard to finance, if the sustainable provision of financing can be guaranteed for the volunteer program, the volunteer services may be better promoted, and the sustainable development of the program may be ensured. To achieve this goal, the leader should have some money-generating activities that will ensure adequate finance to implement their project [23]. A volunteer team that only provides volunteer services can consider transforming into a private non-profit enterprise organization to improve financial sustainability and better promote team and volunteer services. However, in this way, its product and service positioning and organization structure need to be comprehensively considered to realize its "self-hematopoietic" function.

## Limitations

A limitation of our study is that qualitative data on specific factors affecting volunteers' turnover were not available in our study to provide in-depth information. In addition, the quality of the volunteer care was not rated in the present study. A qualitative study concerning the quality of volunteer care needs to be conducted to explore the quality perception from the perspective of the seniors receiving care and their family. Finally, a self-rated formative evaluation was used in this study. There may be a Hawthorne effect when the volunteer team initiator assessed the activity impact. Triangulation of data collection sources and analysis methods would enhance the validity of the study.

## Conclusion

High rate of trained volunteers' dropout in the present program should be brought to the forefront of the program design and to the policymaker's attention. The characteristics of the retained trained volunteer leaders provide a useful reference for the recruitment of trainees in the future. Frailty care may need more training by the volunteer service provider.

Regarding the factors influencing the national volunteer pilot program performance, team climate and atmosphere, financial sustainability, volunteer role identity and attitude toward helping others have significant influences on the performance of the volunteer program. Our recommendations are that when volunteers are provided with a better team climate and atmosphere, such as close interactions between team members, incentives, support and commitment from leaders, and when the sustainability of finance can be ensured, volunteer program performance will be enhanced. Furthermore, appropriate volunteers who incorporate the volunteer role into their self-concept and have an attitude toward helping others are also necessary to enhance program performance.

## Supporting information

**S1 File. Study questionnaire.**
(DOC)

## Acknowledgments

We would express our gratitude to the experts who provided some suggestions to revise the online questionnaire in our pilot study. We are also grateful to all the study participants who were willing to provide precious information for this study. We would also acknowledge the volunteers who devoted themselves to this volunteer program with no payment.

## Author Contributions

**Conceptualization:** Hong-Li Chen, Xian-Wen Li.

**Data curation:** Hong-Li Chen, Xian-Wen Li.

**Formal analysis:** Yu Zhang, Xian-Wen Li.

**Investigation:** Pei Chen, Dao-Xiang Cheng.

**Methodology:** Ying Xing, Yuan-Yuan Guan.

**Project administration:** Xian-Wen Li.

**Resources:** Dao-Xiang Cheng.

**Supervision:** Yuan-Yuan Guan.

**Writing – original draft:** Hong-Li Chen, Pei Chen, Xian-Wen Li.

**Writing – review & editing:** Hong-Li Chen, Pei Chen, Yu Zhang, Ying Xing, Xian-Wen Li.

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
