## [Decision Letter · Decision Letter 0]

11 Jun 2020

PONE-D-20-08937

Involvement of volunteers in community senior care in China: Evaluation of one national pilot program, the Senior Care Volunteers Training Program

PLOS ONE

Dear Dr. Xianwen Li,

Thank you for submitting your manuscript to PLOS ONE. After careful consideration, we feel that it has merit but does not fully meet PLOS ONE’s publication criteria as it currently stands. Therefore, we invite you to submit a revised version of the manuscript that addresses the points raised during the review process.

We look forward to receiving your revised manuscript.

Kind regards,

Sharon Mary Brownie

Academic Editor

PLOS ONE

Additional Editor Comments:

Please carefully consider and respond to the reviewer recommendations. Please construct a file with you response to each suggestion and submit this with your revised manuscript.

Reviewers' comments:

Reviewer's Responses to Questions

**Comments to the Author**

1. Is the manuscript technically sound, and do the data support the conclusions?

Reviewer #1: Partly

Reviewer #2: Yes

2. Has the statistical analysis been performed appropriately and rigorously? 

Reviewer #1: Yes

Reviewer #2: Yes

3. Have the authors made all data underlying the findings in their manuscript fully available?

Reviewer #1: Yes

Reviewer #2: Yes

4. Is the manuscript presented in an intelligible fashion and written in standard English?

Reviewer #1: Yes

Reviewer #2: Yes

5. Review Comments to the Author

Reviewer #1: Dear Authors,

This paper reports important findings related to the SCVTP pilot project. I have listed a few comments to further improve the paper.

1. Study title: You may want to simplify the title to be more straight forward and appealing to the reader. My suggestion: Volunteers' Involvement and reasons for Their Retainment in the Senior Care Volunteers Training Program in Jiangsu, China

2. Please provide further information on sampling method, study outcome and the methods used to measure programme performance in the abstract.

3. Please provide more context on the volunteers who participated in this program in the Introduction section of the article. Who ware they? How were they recruited? Any payment/honorarium given? what are their roles?

4. The methodology employed to measure the programme performance is not clear. Further explanation on VPPE score is important to provide clarity on the study outcome. What are the dependent variables? How do you measure retainment of participants/ how many times did you do the data collection? How do you sample the participants? What is your sampling frame? How do you ensure the validity and reliability of the instrument used in this study?

5. The results and conclusion presented are not in line with the objectives of the article. Please review.

Reviewer #2: The study report useful findings on community volunteers in China. It highlights areas of strength and weaknesses of the volunteer programme. Using various validated tools, the authors assess the performance of the volunteers and factors associated with the performance score such as team climate and atmosphere, role identify, attitude towards helping other and finances. The findings would be useful to policymakers in China in improving the volunteer involvement in community seniors. While the manuscript provide useful information, the authors need to address the following major and minor areas:

Major

1. The introduction/literature review should be shortened and reviewed for clarity. Most information under literature review can easily be summarised into a paragraph which that then be included in the introduction.

2. Revise the abstract to make it short, clear and concise. The results in the abstract especially those on multiple regression review in line with other comments below.

3. The study aims should also be revised to make them clear.

Minor

1. Line 75-76 should be revised to make it clearer

2. Line 154-155: The part of the statement from “..addressing….performance.” should be deleted

3. Line 157: Does “The person..” mean “Everyone”?

4. The authors should indicate percentages using brackets

5. Line 257-259: The statement is based on the data presented in Table 4 is unclear. The table is also unclear especially the numbers in brackets.

6. The authors should use standard statistical abbreviation/notations e.g. M does not represent mean and is confusing when used.

7. Table 5 includes a column with possible range. What is possible range and how different is it from range?

8. Line 273 should be revised for clarity

9. Table 7: The p-values should be presented in a better manner e.g. using asterisks (*/**) to avoid obscuring the correlation coefficients

10. Delete line 287

11. Line 287-290 is a repeats information already in Table 8. The authors should report their findings and not repeat information already in the table. For example what does R2 or the various βs mean?

12. Table 8 should also be revised to communicate the data effectively. For example, for what values (β, P, B, SE) are the 95% CI for?

13. I suggest that the authors combine table 3 and 4 and include annual percentages for values in table 3.

14. Line 284: VPPQ should be revised to VPPE

15. The manuscript would also benefit from further English editing

---

## [Author Response · Author response to Decision Letter 0]

11 Jul 2020

Reviewer #1:

1. Study title: You may want to simplify the title to be more straight forward and appealing to the reader. My suggestion: Volunteers' Involvement and reasons for Their Retainment in the Senior Care Volunteers Training Program in Jiangsu, China.

Response: Thank you very much for your suggestion about the title. We did want to simplify the title to be more straight forward. However, we may not handle it well. Considering your above suggestion and the contents of the article (factors that influencing program performance), we have revised the original title to “Retention of volunteers and factors influencing program performance of the Senior Care Volunteers Training Program in Jiangsu, China.”

We deleted the original title and highlighted revised title in yellow in the file 'Revised Manuscript with Track Changes'.

2. Please provide further information on sampling method, study outcome and the methods used to measure programme performance in the abstract.

Response: Thank you very much for your comments about the Abstract section. We indeed missed these details in the abstract. In this study, all the volunteer group leaders (n=623) who attended SCVTP in the year of 2017, 2018 and 2019 were recruited. Previous research (Black, 2010) has shown that purposive sampling method may prove to be effective when only limited numbers of people can serve as primary data sources due to the nature of research design and aims. The present study used purposive sampling method to select volunteer group leaders who met the criteria, which helped in simplifying the selection of the subject of the investigation. Study outcomes included dropout rate of the trained volunteer group leaders, and the program performance of SCVTP. The method used to measure program performance was via VPPE questionnaire.

We have added and highlighted above information in yellow in the file 'Revised Manuscript with Track Changes'.

Reference

Black, K. Business statistics: Contemporary decision making. 6th ed. Jefferson City: John Wiley; 2010.

3. Please provide more context on the volunteers who participated in this program in the Introduction section of the article. Who were they? How were they recruited? Any payment/honorarium given? what are their roles?

Response: Thanks very much for the suggestions about the Introduction section. Volunteer group leaders with at least two years of experience in senior care were recommended by the local Red Cross Society. There was no payment for them. After the training, if volunteer group leaders pass the test, they would be afforded a certificate and the Red Cross Society of Jiangsu Province would supply twenty thousand Chinese Yuan to encourage them to launch a new volunteer group or to develop an original group taking care for local community senior.

We have added these explanations in the introduction section and highlighted them in yellow in the file 'Revised Manuscript with Track Changes'.

4. The methodology employed to measure the programme performance is not clear. Further explanation on VPPE score is important to provide clarity on the study outcome. What are the dependent variables? How do you measure retainment of participants/ how many times did you do the data collection? How do you sample the participants? What is your sampling frame? How do you ensure the validity and reliability of the instrument used in this study?

Response: Thank you for your comments. We have given further explanation of the VPPE meaning and the dependent variable in the Instruments section. For the regression model, the VPPE score was the dependent variable. The higher weighted VPPE score, the better program performance was. As for the way to measure retainment of participants, sample the participants and sample frame, we did miss these details in the manuscript. We have added related statement in the study participants and dropout section and data collection section, respectively. In this study, all the volunteer group leaders who attended SCVTP developed in the year of 2017, 2018 and 2019 were recruited. The sampling frame consisted of a roster of all the trained volunteer group leaders. The study used purposive sampling to select participants who met the criteria, which has been explained in the above response. The criteria used to measure retainment of participants are as follows: (1) The trained volunteer group leaders who did not respond to this study were defined as one kind of dropout. (2) Even though they responded to this investigate but did not launch a volunteer group program caring for local community senior were defined as the dropout. (3) The respondents who did not meet “response time” and “self-reported diligence” criteria were also defined as the dropout (Bowling NA, 2016; DeSimone JA, 2017). The data was collected only once, and we sent three reminders to volunteer group leaders about the online questionnaire. Cronbach’s α was used to ensure the psychometric property of the instrument (The Cronbach’s α of VPPE questionnaire have been added). Concordance coefficients and variation coefficients were also the way to ensure the psychometric property of VPPE when we framed this tool. 

These contents have been highlighted in yellow in the file 'Revised Manuscript with Track Changes'.

References

Bowling NA, Huang JL, Bragg CB, Khazon S, Liu M, Blackmore CE. Who cares and who is careless? Insufficient effort responding as a reflection of respondent personality. J Pers Soc Psychol. 2016; 111(2):218-29. https://doi.org/10.1037/pspp0000085 PMID: 26927958

DeSimone JA, Harms PD, DeSimone AJ. Best practice recommendations for data screening. Journal of Organizational Behavior. 2017,36(12):171-181. https://doi.org/10.1002/job.1962

5. The results and conclusion presented are not in line with the objectives of the article. Please review.

Response: Thank you very much for your comments about this question. We have revised some statements so that the results and conclusion are in line with the objectives of the article. The objectives, results and conclusion are as follows:

Objectives:

(1) To investigate the dropout rate of trained volunteer group leaders.

(2) To describe the characteristics of the retained volunteer group leaders and the activities. 

(3)To explore the factors influencing program performance. 

Results:

(1) About 67.9%, 53.7%, and 30.0% of the trained volunteer group leaders dropped out of the program in 2017, 2018, and 2019, respectively. 

(2) The retained trained volunteer group leaders were more likely to be females (84.7%), those in excellent health (75.2%) and those with a bachelor’s degree or above (87.6%). Less attention has been paid to frailty care (n=76) than other volunteer caring activities. 

(3) VRI (β=0.118, p=0.017), AHO (β=0.134, p=0.021), TCA (β=0.459, p<0.001), and financial sustainability (β=0.179, p<0.001) affected the SCVTP’s performance significantly (adjusted R2 =0.356). 

Conclusion:

(1) High rate of trained volunteer group leaders’ dropout should be brought to the policymaker’s attention.

(2) The characteristics of the retained trained volunteer group leaders provide a useful reference for the recruitment of trainees in the future. Frailty care may need more training by the volunteer service provider. 

(3)In order to enhance volunteer program performance, a better team climate and atmosphere, financial sustainability, and volunteers with appropriate attitude and role identity are necessary.

Reviewer #2:

Major

1. The introduction/literature review should be shortened and reviewed for clarity. Most information under literature review can easily be summarised into a paragraph which that then be included in the introduction.

Response: Thank you very much for the suggestion about introduction/literature review section. The literature review section really seemed to be too long. We have summarized these into a paragraph and included them in the introduction section. 

These contents have been highlighted in blue in the file ‘Revised Manuscript with Track Changes’.

2. Revise the abstract to make it short, clear and concise. The results in the abstract especially those on multiple regression review in line with other comments below.

Response: Thanks for your comments. We have tried our best to revise the abstract to make it short, clear and concise. 

The content we revised has been highlighted in red in the file ‘Revised Manuscript with Track Changes.

3. The study aims should also be revised to make them clear.

Response: Thank you very much for your suggestion. The aims in the abstract have been revised and included in the background section. The aims in the introduction section were stated in the form of ‘the first aim, the second, the third…’ to make them clear. The aims of the article are as follows: (1) To investigate the dropout rate of trained volunteer group leaders. (2) To describe the characteristics of the retained volunteer group leaders and the activities. (3) To explore the factors influencing program performance.

The content we revised has been highlighted in red in the file ‘Revised Manuscript with Track Changes’.

Minor

1. Line 75-76 should be revised to make it clearer.

Response: Thank you very much for your suggestions about writing error and statement details. We really did not deal with these details well, and are so sorry about this. Line 75-76 have been revised to be clearer and highlighted in blue in the file ‘Revised Manuscript with Track Changes’. (Now: line 78-79)

2. Line 154-155: The part of the statement from “..addressing….performance.” should be deleted.

Response: We have deleted the part of statement from “...addressing…performance.”

3. Line 157: Does “The person..” mean “Everyone”?

Response: We have revised it to make it clear. “The person..” means “sample”. In the study participants and dropout section, we have added more details about population, sample method and sample frame to make this section clearer. (Now: line 108-115)

4. The authors should indicate percentages using brackets.

Response: We have used brackets to indicate percentages. 

The content we revised has been highlighted in blue in the file ‘Revised Manuscript with Track Changes’.

5. Line 257-259: The statement is based on the data presented in Table 4 is unclear. The table is also unclear especially the numbers in brackets.

Response: The options in the “Types of respite care/activities” is multiple choice, the respondents may choose one or more types of care if they did conduct these types of care. All the numbers/percentages cannot be added up to 100%. So, we have to say that the numbers/percentages in brackets seemed to be unclear (Now: table 3), added the expression in the instruments section (Now: line 127-128). We have revised the statement based on table 4 (Now: table3) to make it clear. (Now: line 215-217) 

The content we revised has been highlighted in blue in the file ‘Revised Manuscript with Track Changes’. 

6. The authors should use standard statistical abbreviation/notations e.g. M does not represent mean and is confusing when used.

Response: We have revised and used standard statistical abbreviation/notations.

The content we revised has been highlighted in blue in the file ‘Revised Manuscript with Track Changes’.

7. Table 5 includes a column with possible range. What is possible range and how different is it from range?

Response: ‘Possible range’ refers to minimum to maximum score of the scale，while ‘range’ means to actual minimum to maximum score that respondents answered.(Now: table 4)

The content we revised has been highlighted in blue in the file ‘Revised Manuscript with Track Changes’.

8. Line 273 should be revised for clarity.

Response: We have revised to make it clear. 

The content we revised has been highlighted in blue in the file ‘Revised Manuscript with Track Changes’. (Now: line 231-233)

9. Table 7: The p-values should be presented in a better manner e.g. using asterisks (*/**) to avoid obscuring the correlation coefficients.

Response: We have revised the way p-values presented which used asterisks (*/**). 

The content we revised has been highlighted in blue in the file ‘Revised Manuscript with Track Changes’. (Now: table 6)

10. Delete line 287

Response: We have deleted original line 287.

11. Line 287-290 is a repeats information already in Table 8. The authors should report their findings and not repeat information already in the table. For example what does R2 or the various βs mean?

Response: We have given more information about the meaning of R2 and β. The meaning of β is that a unit change in VRI, AHO, TCA, finance increased program performance by 0.155, 0.346, 1.197, 0.478 units, respectively. The adjusted R2 of 0.356 indicated the power of model for prediction its significance at 0.05 level of probability and revealed that 35.6 percent of variance in participation could be explained by four aforementioned variables (Now: line 247-250). 

The content we revised has been highlighted in blue in the file ‘Revised Manuscript with Track Changes’.

12. Table 8 should also be revised to communicate the data effectively. For example, for what values (β, P, B, SE) are the 95% CI for?

Response: We have added that what values (β, P, B, SE) are for. (B: unstandardized regression coefficient; S.E.: Std.Error; β: standardized regression coefficient; p: p-value; CI: confidence interval.)

The content we revised has been highlighted in blue under Table 8 (Now: Table 7) in the file ‘Revised Manuscript with Track Changes’.

13. I suggest that the authors combine table 3 and 4 and include annual percentages for values in table 3.

Response: We have combined table 3 and 4 and included annual percentages for values in table 3.

14. Line 284: VPPQ should be revised to VPPE.

Response: We are really sorry for this written error. VPPQ has been revised to VPPE. 

The content we revised has been highlighted in red in the file ‘Revised Manuscript with Track Changes’.

15. The manuscript would also benefit from further English editing.

Response: We have used the Standard Editing service of American Journal Experts to improve the manuscript.

---

## [Decision Letter · Decision Letter 1]

27 Jul 2020

Retention of volunteers and factors influencing program performance of the Senior Care Volunteers Training Program in Jiangsu, China

PONE-D-20-08937R1

Dear Dr.Xianwen Li 

We’re pleased to inform you that your manuscript has been judged scientifically suitable for publication and will be formally accepted for publication once it meets all outstanding technical requirements.

Kind regards,

Sharon Mary Brownie

Academic Editor

PLOS ONE

Additional Editor Comments (optional):

Appropriate revisions have been completed in response to reviewer recommendations.

Reviewer's Responses to Questions

**Comments to the Author**

1. If the authors have adequately addressed your comments raised in a previous round of review and you feel that this manuscript is now acceptable for publication, you may indicate that here to bypass the “Comments to the Author” section, enter your conflict of interest statement in the “Confidential to Editor” section, and submit your "Accept" recommendation.

Reviewer #2: All comments have been addressed

2. Is the manuscript technically sound, and do the data support the conclusions?

Reviewer #2: Yes

3. Has the statistical analysis been performed appropriately and rigorously? 

Reviewer #2: Yes

4. Have the authors made all data underlying the findings in their manuscript fully available?

Reviewer #2: Yes

5. Is the manuscript presented in an intelligible fashion and written in standard English?

Reviewer #2: Yes

6. Review Comments to the Author

Reviewer #2: The authors have satisfactorily addressed all the reviewers comments. Some minor typographical and grammatical errors need to be revised to improve clarity of some statements e.g.Line 66, Lines 100-102, Line 128, Line 201 and Line 328 ("made an assessment of" to "assessed"). Also, under education consider using "undergraduate" rather than "bachelor".

7. PLOS authors have the option to publish the peer review history of their article (what does this mean?). If published, this will include your full peer review and any attached files.

Reviewer #2: **Yes: **Samwel Maina Gatimu